# Readmission Events Following EGD for Upper Gastrointestinal Bleed: An Analysis Using the National Readmission Database

**DOI:** 10.3390/medsci13020045

**Published:** 2025-04-20

**Authors:** Vignesh Krishnan Nagesh, Vivek Joseph Varughese, Jaber Musalli, Gomathy Aarthy Nageswaran, Erin Russell, Susan Anne Feldman, Simcha Weissman, Adam Atoot

**Affiliations:** 1Hackensack Palisades Medical Center, 7600 River Rd, North Bergen, NJ 07047, USA; musallijaber@gmail.com (J.M.); erine.russell@hmhn.org (E.R.); simchaweissman@gmail.com (S.W.); adamatoot.md@gmail.com (A.A.); 2Prisma Health, University of South Carolina, 2 Med Park, Richland, Columbia, SC 29203, USA; susan.feldman@prismahealth.org; 3Internal Medicine, University of Arkansas for Medical Sciences (UAMS), Little Rock, AR 72205, USA; gomathyaarthy@gmail.com

**Keywords:** upper gastrointestinal bleeding (UGIB), esophagogastroduodenoscopy (EGD), national readmission database (NRD), mortality, variceal bleed

## Abstract

**Background:** Upper Gastrointestinal Bleed (UGIB) is a common and potentially life-threatening condition with an annual incidence of 80–150 per 100,000 individuals and a mortality rate of 2–10%. Esophagogastroduodenoscopy (EGD) is the gold standard for both diagnosis and treatment, but post-discharge outcomes, including readmissions, remain underexplored. **Methods:** This study utilized the 2021 National Readmission Database (NRD) to analyze 30-day readmission rates following EGD for UGIB. Adult patients (≥18 years) admitted for UGIB and undergoing EGD were included; those who died during the index hospitalization were excluded. Demographic, clinical, and socioeconomic factors associated with readmission were examined using multivariate logistic regression. **Results:** Among 34,257 patients admitted for UGIB and undergoing EGD, 11,088 (32.4%) were readmitted within 30 days, with 5423 (49%) due to recurrent UGIB. Readmitted patients had a higher mean age (68.46 vs. 67.63 years) and greater prevalence of cirrhosis (16.71% vs. 13.84%). Hospital resource utilization was significantly higher among readmissions, with increased total hospital charges (USD 82,544.82 vs. USD 61,521.17) and longer hospital stays (5.38 vs. 4.97 days). Mortality was lower among readmitted patients (1.46% vs. 3.53%). Multivariate analysis identified cirrhosis (OR 7.20, 95% CI: 6.45–8.02), untreated H. pylori infection (OR 3.43, 95% CI: 2.15–4.30), atrial fibrillation (OR 1.52, 95% CI: 1.36–1.69), and chronic antithrombotic therapy (OR 1.63, 95% CI: 1.41–1.89) as significant predictors of recurrent UGIB readmission. Lower socioeconomic status was also associated with increased readmission risk (OR 1.15, 95% CI: 1.05–1.25). **Conclusions:** Readmission following EGD for UGIB is common and driven primarily by recurrent bleeding. Cirrhosis, untreated H. pylori infection, atrial fibrillation, and chronic anticoagulation therapy are key risk factors. These findings highlight the need for targeted interventions, including improved post-discharge management and optimization of anticoagulation strategies, to reduce readmission rates and improve patient outcomes.

## 1. Introduction

Upper Gastrointestinal Bleeding (UGIB) refers to any bleeding that occurs in the gastrointestinal (GI) tract from the mouth proximal to the ligament of Treitz [1]. Common causes of UGIB include gastric ulcers, gastric or duodenal erosions, arteriovenous malformations (AVM), erosive esophagitis, esophageal variceal bleed, Mallory–Weiss Tears (MWTs), and hemobilia [2]. UGIB occurs in 80 to 150 per 100,000 in the general population and has an estimated mortality of 2–10%, making it an important condition to triage and appropriately manage in the acute setting [3]. Presenting symptoms include coffee ground emesis, fresh blood vomiting, hematochezia, or melena, along with systemic symptoms like syncope, fatigue, and weakness from blood loss, leading to hemoglobin drop and eventually anemia [4,5,6].

Initial management of acute UGIB focuses on hemodynamic stabilization and determination of the source of bleeding [2,3]. When determining the source of bleeding, esophagogastroduodenoscopy (EGD) is the diagnostic modality of choice because it enables the visualization of the upper GI tract from the mouth to the second portion of the duodenum and the visualization of the ampulla of Vater [2]. It has been reported that EGD can identify the culprit lesion in more than 80% of patients [7]. EGD also serves a dual purpose in the setting of active bleeding or high-risk stigmata of recent bleeding by allowing for localized treatment through injection of medication, variceal ligation, use of thermal probes to cauterize active bleeding sites, or application of clips to promote hemostasis [2,3,5].

There are also associated risks of EGD in the setting of acute UGIB. Adverse effects have been reported to occur in 0.01% to 0.4% of overall patients who undergo diagnostic EGD, with rates ranging from less than 0.5% for patients undergoing endoscopic non-variceal hemostasis to rates of 35% to 78% for those undergoing endoscopic variceal sclerotherapy [8]. Reported adverse events include bleeding, perforation, infection, aspiration, and cardiopulmonary events associated with the sedation [8]. There is limited information available on adverse events other than recurrent variceal bleeding occurring after discharge or at a later time point [8].

The timing of EGD is recommended by an international consensus group to be within 24 h due to reduced length of stay for low- and high-risk patients and reduced need for surgery in elderly patients [9] A recent study found that there was no difference in 30-day mortality for high-risk patients between those undergoing EGD within 6 h or between 6 and 24 h [10]. Notably, this same study found the 30-day mortality amongst high risk patients to be between 6.6% to 8.9% [10]. This affirms the importance of identifying factors associated with poor outcomes such as readmissions. Several risk stratification systems exist for GI bleeding, with the Glasgow-Blatchford, AIMS65, and Rockall scores used for UGIB and the Oakland score for LGIB. The Glasgow-Blatchford score is the preferred tool for emergency assessment, identifying low-risk patients eligible for outpatient care, though debate exists on adjusting the risk threshold. In the ICU, the Rockall and AIMS65 scores assess UGIB mortality risk, while the Oakland score predicts safe discharge in LGIB based on factors like vitals, hemoglobin levels, and clinical history [11,12,13,14].

A meta-analysis of studies reporting readmission rates for patients following a UGIB found the 30-day all-cause readmission rate to be 17.4%, with variceal UGIB resulting in a higher rate. One-third of readmissions were due to recurrent UGIB, but the reason for the remainder of the events was not reported [15]. For variceal UGIB, the top diagnoses found to be associated with readmission were cirrhosis with ascites, GI hemorrhage, variceal hemorrhage, sepsis, and hepatic failure [16]. For non-variceal UGIB, a study of a US database found the top diagnoses associated with readmission to be GI tract hemorrhage, septicemia, gastric ulcer with hemorrhage, duodenal ulcer with hemorrhage, and angiodysplasia of the stomach and duodenum with hemorrhage [17]. A study in Sweden found rebleeding to be the primary cause for readmission, followed by bacterial infections and cardiovascular events [18].

None of these studies reported readmission data specific to EGD for UGIB. This analysis set out to identify the readmission events associated with EGD for UGIB using a nationwide database.

## 2. Methods

The National Readmission Database (NRD) 2021 was used for the analysis. Using the ICD 10 diagnosis codes K920, K921, and K922, index events recorded between the months of January and November of 2021 with a main diagnosis of Upper Gastrointestinal Bleed (UGIB) were selected. The PCS 10 code ODJ08** was used to select main admissions for UGIB that underwent esophagogastroduodenoscopy (EGD). Age > 18 and admissions that did not die during the initial admission were used as further inclusion criteria to select the index admission population used to study the 30-day readmission events. Entries with no recorded length of stay, as well as entries lacking unique patient identifier numbers (nrc_vsit_link) and admission identifiers for individual admissions (key_nrd), were excluded while selecting the index population. Thirty-day readmission events were analyzed from the day of discharge of the index admission event.

Population characteristics of the index admission events and readmission events for recurrent gastrointestinal (GI) bleed were analyzed. CMR diagnosis codes were used to select admissions with cirrhosis and alcohol abuse, while ICD 10 codes Z920 and Z921 were used to select patients with chronic use of long-term anticoagulation and antiplatelet therapy. While determining the presence of comorbid pathological conditions, IBD (ICD 10 code K500**), untreated H. Pylori infections (ICD 10 code B9681), and atrial fibrillation (CMR code) were used. Mortality among index admissions and repeat GI bleed admissions were studied. Weighted national averages using nrd_stratum and hospital identifiers were used to select the means for total admission and readmission causes. Mean length of hospital stay was also compared between the index admissions and rebleeds to assess the healthcare resource utilization burden. Causes of readmission other than recurrent GI bleed in the 30 days following discharge for the initial admission were analyzed and reported.

To study the association between repeat GI bleeds in 30 days following discharge and patient comorbidities, multivariate logistic regression (probit model) was used. Age, sex, alcohol abuse, and chronic anticoagulation and antiplatelet therapy were used as confounders in the regression analysis. A two-tailed *p* value was used to determine the statistical significance of the associations. To assess the association of socioeconomic factors with readmissions for GI bleed, the variable zipinc_qrtl, which notes the monthly income quartile of the admission population, was used to determine admissions belonging to the lower quartile of the monthly income category.

## 3. Results

A total of 92,615 admissions for GI bleed were recorded. Using the readmissions database, we identified 34,257 index event admissions for UGIB that underwent EGD during the admission between the months of January and November of 2021. Among the recorded admissions, 11,088 patients experienced readmission within 30 days of discharge, totaling 21,137 readmission events within this period. Specifically, 5423 of these readmissions occurred within 30 days following an initial index admission for GI bleeding. This shown in Figure 1.

The population and social characteristics of the index UGIB admissions and the readmissions for rebleed were analyzed. The results are summarized in Table 1 and Figure 1.

Mortality trends among index UGIB admissions, as well as among admissions related to rebleeding in the 30 days following discharge, were analyzed. Healthcare resource utilization was analyzed in terms of mean total hospital charges and mean lengths of hospital stay. The results are summarized in Table 2.

The causes of 30-day readmissions following discharge for index admissions for UGIB that underwent EGD were also analyzed, and the results are summarized in Figure 2.

The socioeconomic distribution of the population with recurrent GI bleeds in 30 days following EGD for index UGIB events was analyzed based on monthly quarterly income. The results are summarized in Figure 3.

The prevalence of comorbid disease conditions, namely inflammatory bowel disease (IBD), untreated H. pylori infections, atrial fibrillation (afib), and cirrhosis, was analyzed in index UGIB patients and in the admissions pool with 30-day repeat GI bleed after EGD after the initial bleeding event. The results are summarized in Table 3 and Figure 4, respectively.

In analyzing the association of comorbid conditions with the occurrence of 30-day repeat GI bleed after EGD for initial UGIB, multivariate logistic regression was employed. Age, sex, and hospital region and size were used in the regression analysis to account for confounding. To check the associations between patient comorbid conditions, alcohol abuse, cirrhosis, anticoagulation, and antiplatelet use, as well as monthly income, were used in the regression analysis. A two-tailed *p*-value < 0.05 was used to determine statistical significance. The results are summarized in Table 4 and Figure 5, respectively.

## 4. Discussion

In this study, we analyzed readmission events following esophagogastroduodenoscopy (EGD) for Upper Gastrointestinal Bleed (UGIB) using the National Readmission Database (NRD). Among 34,257 index admissions for UGIB that underwent EGD, 11,088 patients (32.4%) experienced readmission within 30 days, with 5423 cases (49%) due to recurrent gastrointestinal bleeding (GIB). Notably, readmitted patients also had higher hospital charges and longer lengths of stay, while the overall mortality rate was lower among readmissions (1.46%) compared to index admissions.

Our analysis of the NRD further revealed that patients readmitted within 30 days following EGD for UGIB exhibited notable differences compared to those that were not readmitted. Readmitted patients had a slightly higher mean age (68.46 vs. 67.63 years) and a higher prevalence of cirrhosis (16.71% vs. 13.84%). Hospital resource utilization was significantly higher among readmissions, with greater mean total hospital charges (USD 82,544.82 vs. USD 61,521.17) and longer lengths of stay (5.38 vs. 4.97 days). Despite this increased utilization, the lower all-cause mortality among readmitted patients may be attributable to the early identification and management of recurrent bleeds in these high-risk individuals.

Multivariate logistic regression revealed that several factors were significant predictors of 30-day recurrent GI bleeding. In particular, cirrhosis (adjusted OR 7.20, 95% CI: 6.45–8.02), untreated H. pylori infection (adjusted OR 3.43, 95% CI: 2.15–4.30), atrial fibrillation (adjusted OR 1.52, 95% CI: 1.36–1.69), and chronic antithrombotic therapy (adjusted OR 1.63, 95% CI: 1.41–1.89) emerged as significant predictors. Interestingly, neither inflammatory bowel disease (IBD) nor NSAID use was significantly associated with an increased risk of readmission. Moreover, socioeconomic disparities played a modest role, with patients in the lowest income quartile exhibiting a 15% higher odds of 30-day readmission (adjusted OR 1.15, 95% CI: 1.05–1.25). These findings underscore the complexity of comorbidities and social determinants of health in influencing post-discharge outcomes, highlighting the importance of comprehensive discharge planning and targeted interventions for high-risk populations.

Prior research by Garg et al. using the National Inpatient Sample (NIS) demonstrated that early EGD (within 24 h of admission) is associated with significantly lower mortality compared to delayed or no EGD [19]. However, due to the cross-sectional nature of NIS data, such studies were unable to evaluate readmission rates or associated post-discharge morbidity and mortality. Our study provides additional insights by identifying atrial fibrillation and chronic anticoagulation therapy as significant predictors of 30-day readmissions, a finding that aligns with the prospective observational study by Sengupta et al. [20].

Further supporting our observations, a retrospective study by Daðadóttir et al. found higher mortality rates among cirrhotic patients experiencing UGIB compared to non-cirrhotic patients, emphasizing the need for targeted interventions in this high-risk population [21]. In contrast, our analysis did not find an association between 30-day recurrent UGIB and IBD or NSAID use. This observation is corroborated by a retrospective cohort study by Arjonilla et al. [22], which identified dual antiplatelet therapy, anemia, and specific endoscopic findings, rather than IBD or NSAID use, as key risk factors for recurrent UGIB. Similarly, Tapaskar et al. [23] demonstrated that while resuming anticoagulation post-UGIB can reduce thromboembolic events and improve mortality, it also increases the odds of recurrent bleeding by 64%, thereby necessitating a nuanced risk–benefit analysis prior to discharge. Antiplatelet therapy is essential for the secondary prevention of cardiovascular events but increases the risk of GI bleeding. Early endoscopy and PPI co-therapy are recommended to minimize bleeding risk [24].

Esophageal variceal bleeding secondary to liver cirrhosis is one of the most common causes of UGIB. Pharmacological prophylaxis of variceal bleeds includes treatment with non-selective beta blockers. Endoscopic therapy for bleeding esophageal varices includes endoscopic band ligation (EBL) and injection sclerotherapy, with EBL as the preferred approach due to fewer complications. A study comparing EBL alone vs. EBL with sclerotherapy found that the combination significantly reduced re-bleeding recurrence. Argon plasma coagulation (APC), used adjunctively with EBL or sclerotherapy, has also been shown to lower variceal recurrence, rebleeding, and mortality. For persistent or recurrent variceal bleeding unresponsive to initial therapy, rescue transjugular intrahepatic portosystemic shunt (TIPS) within eight hours improves survival outcomes [25,26,27,28,29,30,31].

In addition to these clinical factors, hospital characteristics and access to advanced specialized care also influence outcomes. Siddique et al. [32] have shown that these factors can affect both mortality and length of stay, and our findings suggest that broader socioeconomic disparities—such as limited access to post-discharge follow-up, financial barriers to medication adherence, and overall healthcare resource availability—may further contribute to the risk of readmission.

Recent studies have added further depth to our understanding of these issues. For instance, Bilicki et al. [33] reported that structured discharge planning and early outpatient follow-up significantly reduce readmission rates in UGIB patients, while Cho et al. [34] observed that rapid endoscopic intervention and tailored risk stratification are critical to reducing both in-hospital mortality and subsequent readmissions. Moreover, Viderman et al. [35] emphasized the potential role of telemedicine in the post-discharge management of high-risk UGIB patients, suggesting that remote monitoring may be an effective tool in mitigating recurrent bleeding events.

Despite these valuable insights, our study has several limitations. The NRD’s reliance on administrative coding data introduces the potential for misclassification bias, and its lack of detailed clinical data—such as endoscopic findings, hemodynamic instability at presentation, or adherence to guideline-directed therapy—limits the depth of our analysis. Furthermore, the NRD’s inability to track patients beyond a single year precludes the assessment of long-term outcomes, and the lack of state-specific analyses prevents regional comparisons and limits the examination of geographic disparities.

In summary, our study demonstrates that atrial fibrillation, chronic anticoagulation, and cirrhosis are key predictors of 30-day readmission for recurrent UGIB. These findings emphasize the need for enhanced post-discharge monitoring and tailored risk stratification strategies, particularly among patients facing socioeconomic disparities. While our study sheds light on important predictors of readmission following EGD for UGIB, several questions remain unanswered. Future research should focus on structured post-discharge care—including early outpatient visits, telemedicine, and multidisciplinary discharge planning—to evaluate their impact on reducing readmissions. Additionally, incorporating detailed endoscopic findings and procedural data could help identify specific lesion types or techniques associated with recurrence. Finally, developing evidence-based guidelines for resuming anticoagulation through individualized bleeding and thrombotic risk assessments, as well as addressing socioeconomic and healthcare access disparities through qualitative research and policy interventions, will be crucial in reducing readmission inequities and improving long-term patient outcomes.

## 5. Conclusions

This study analyzes 30-day readmissions after EGD for UGIB using the NRD. Nearly one-third of patients were readmitted, with half due to recurrent GIB. Readmitted patients had higher rates of cirrhosis, atrial fibrillation, and chronic anticoagulation, highlighting the impact of comorbidities. Readmission was associated with greater healthcare utilization, yet lower mortality than index admissions, suggesting effective early intervention. Cirrhosis was the strongest predictor of recurrent GIB, alongside untreated H. pylori, atrial fibrillation, and anticoagulation. Lower income also increased readmission risk, highlighting socioeconomic disparities. Targeted strategies, including enhanced discharge planning, outpatient follow-up, and anticoagulation optimization, are needed to reduce readmissions. Future research should explore interventions to improve long-term outcomes.

## Data Availability

The data presented in this study are openly available in the NRD and are exempt from IRB approval.

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
