# Peer review of "Readmission Events Following EGD for Upper Gastrointestinal Bleed: An Analysis Using the National Readmission Database"

_medsci, 2025, doi:10.3390/medsci13020045_

Round 1

Reviewer 1 Report

Comments and Suggestions for Authors

The authors analyzed the 2021 National Readmission Database (NRD) to assess the 30-day readmission rates and causes of readmission following the index hospitalization due to UGIB. Of 34,257 index admissions for UGIB undergoing EGDS, 32.4% of patients experienced readmission within 30 days, among which 49% were due to recurrent UGIB. The authors found liver cirrhosis, atrial fibrillation, untreated HP infection, and antithrombotic therapy as significant predictors of recurrent UGIB.

  1. This is a study that analyzed a large cohort of patients and is sound and clinically relevant. However, despite proper methodology and analysis of results, the writing of the Manuscript must be improved. References are fine. The English language must be improved.
  2. The main limitation of this study is the lack of information regarding the endoscopic stigmata both in the index and recurrent hospitalization.
  3. The patient inclusion flowchart should be added.
  4. The authors tend to repeat parts of the Methodology section in the Results section, eg:

Lines 115-118 are already stated in the Methods section.

Line 121/122: …between the months of January and 121 November of 2021 (delete this part due to repetition).

Line 122: Admissions under the age of 18…?

Lines 122/123: Admissions under the age of 18, and with no recorded death during 122 the admissions were selected to analyze 30 day events following the discharge: repetition from the methodology section again.

  1. What are the author's key suggestions for clinicians to reduce this high prevalence (30%!) of readmissions, especially in patients with liver cirrhosis and patients taking antithrombotic therapy? Please discuss this part with specific recommendations.
  2. Additionally:

-Line 86 please correct: Age > 18 and admissions that did not die during the initial admissions…

-This part is confusing for readers and should be written more simply and clearly, without repetition:  Of these, 123 11,088 patients had readmission events in the 30 days following discharge. There were a 124 total of 21,137 readmission events in this population in the 30 days following discharge. 125 Of these, there were 5423 recorded readmission events in the 30 days following discharge 126 after initial Index admission for GI bleed.

-Table 3 should be arranged (the text is improperly connected)

Comments on the Quality of English Language

Must be improved. 

Author Response

Hello 

Changes have been made to the manuscript and highlighted .

Reviewer 2 Report

Comments and Suggestions for Authors

Dear Authors

I appreciate the work, which I consider interesting. However, I think it could be improved and I have left my comments in the text.

I do not think it is appropriate to discuss and draw conclusions about variables
not selected in the protocol (NSAIDs); I also believe that some of the variables
chosen are not understandable (IBD).

References will need to be revised to the presentation format required by the editor.

Many regards

Author Response

Hello changes have been made to the manuscript and highlighted in red.

Replies are attached in the file

Round 2

Reviewer 1 Report

Comments and Suggestions for Authors

Patients who experienced readmission should be added to the flowchart and the flowchart should be marked as a Figure/Scheme. 

Author Response

Thank you for this valuable point, changes have been made and the flowchart has been marled as scheme 1 including the readmissions

Reviewer 2 Report

Comments and Suggestions for Authors Dear Authors Thank you for accepting the suggestions. I think the manuscript can be published. Only in Figure 2 is "cirrhosis" in lower case.

Many regards

Author Response

Thank you for reviewing our paper, changes have been made with "cirrhosis' being changed to "Cirrhosis"